# Virtual Biopsy for Diagnosis of Chemotherapy-Associated Liver Injuries and Steatohepatitis: A Combined Radiomic and Clinical Model in Patients with Colorectal Liver Metastases

**DOI:** 10.3390/cancers13123077

**Published:** 2021-06-20

**Authors:** Guido Costa, Lara Cavinato, Chiara Masci, Francesco Fiz, Martina Sollini, Letterio Salvatore Politi, Arturo Chiti, Luca Balzarini, Alessio Aghemo, Luca di Tommaso, Francesca Ieva, Guido Torzilli, Luca Viganò

**Affiliations:** 1Division of Hepatobiliary and General Surgery, Department of Surgery, IRCCS Humanitas Research Hospital, Rozzano, 20189 Milan, Italy; guido.costa@humanitas.it (G.C.); guido.torzilli@hunimed.eu (G.T.); 2Department of Biomedical Sciences, Humanitas University, Pieve Emanuele, 20090 Milan, Italy; martina.sollini@hunimed.eu (M.S.); letterio.politi@hunimed.eu (L.S.P.); arturo.chiti@hunimed.eu (A.C.); alessio.aghemo@hunimed.eu (A.A.); luca.di_tommaso@hunimed.eu (L.d.T.); 3MOX Laboratory, Department of Mathematics, Politecnico di Milano, 20133 Milan, Italy; lara.cavinato@polimi.it (L.C.); chiara.masci@polimi.it (C.M.); 4Department of Nuclear Medicine, IRCCS Humanitas Research Hospital, 20189 Milan, Italy; francesco.fiz@cancercenter.humanitas.it; 5Department of Radiology, IRCCS Humanitas Research Hospital, Rozzano, 20189 Milan, Italy; luca.balzarini@humanitas.it; 6Division of Internal Medicine and Hepatology, Department of Internal Medicine, IRCCS Humanitas Research Hospital, Rozzano, 20189 Milan, Italy; 7Pathology Unit, IRCCS Humanitas Research Hospital, 20189 Milan, Italy; 8CADS—Center for Analysis, Decisions and Society, Human Technopole, 20157 Milan, Italy

**Keywords:** chemotherapy-associated liver injuries, sinusoidal dilatation, nodular regenerative hyperplasia, steatohepatitis, diagnostic imaging, radiomics, textural features, colorectal liver metastases, liver surgery, virtual liver biopsy

## Abstract

**Simple Summary:**

Patients receiving chemotherapy for liver metastases from colorectal cancer may develop liver injuries that impair hepatic function and postoperative outcome. The non-invasive diagnosis of these damages is still an unmet need. Recently, advanced imaging analysis techniques, including the so-called “radiomics”, achieved adequate prediction of pathology data. The present study demonstrated that radiomic analysis of liver parenchyma in combination with clinical and laboratory data improves non-invasive diagnosis of chemotherapy-related liver injuries.

**Abstract:**

Non-invasive diagnosis of chemotherapy-associated liver injuries (CALI) is still an unmet need. The present study aims to elucidate the contribution of radiomics to the diagnosis of sinusoidal dilatation (SinDil), nodular regenerative hyperplasia (NRH), and non-alcoholic steatohepatitis (NASH). Patients undergoing hepatectomy for colorectal metastases after chemotherapy (January 2018-February 2020) were retrospectively analyzed. Radiomic features were extracted from a standardized volume of non-tumoral liver parenchyma outlined in the portal phase of preoperative post-chemotherapy computed tomography. Seventy-eight patients were analyzed: 25 had grade 2–3 SinDil, 27 NRH, and 14 NASH. Three radiomic fingerprints independently predicted SinDil: GLRLM_f3 (OR = 12.25), NGLDM_f1 (OR = 7.77), and GLZLM_f2 (OR = 0.53). Combining clinical, laboratory, and radiomic data, the predictive model had accuracy = 82%, sensitivity = 64%, and specificity = 91% (AUC = 0.87 vs. AUC = 0.77 of the model without radiomics). Three radiomic parameters predicted NRH: conventional_HUQ2 (OR = 0.76), GLZLM_f2 (OR = 0.05), and GLZLM_f3 (OR = 7.97). The combined clinical/laboratory/radiomic model had accuracy = 85%, sensitivity = 81%, and specificity = 86% (AUC = 0.91 vs. AUC = 0.85 without radiomics). NASH was predicted by conventional_HUQ2 (OR = 0.79) with accuracy = 91%, sensitivity = 86%, and specificity = 92% (AUC = 0.93 vs. AUC = 0.83 without radiomics). In the validation set, accuracy was 72%, 71%, and 91% for SinDil, NRH, and NASH. Radiomic analysis of liver parenchyma may provide a signature that, in combination with clinical and laboratory data, improves the diagnosis of CALI.

## 1. Introduction

The combination of chemotherapy and surgery is the standard treatment of patients with colorectal liver metastases (CLM) [1,2]. Systemic chemotherapy prolongs progression-free survival, allows for the selection of the candidates for surgery, and moves some initially unresectable patients to secondary resectability [3,4,5,6,7]. However, systemic treatment may lead to chemotherapy-associated liver injuries (CALI) such as sinusoidal dilatation, nodular regenerative hyperplasia (NRH), and non-alcoholic steatohepatitis (NASH) [8,9,10,11]. Generally, sinusoidal dilatation and NRH are related to oxaliplatin-based chemotherapy [8,10,12], while steatohepatitis is associated with irinotecan-based chemotherapy [9,13]. Steatohepatitis can be caused not only by chemotherapy, but also observed in patients with metabolic disorders [14,15]. This is epidemiologically relevant given the increasing prevalence of such disorders in the general population. CALI are of major interest for liver surgeons because of their association with an increased risk of intraoperative bleeding, postoperative morbidity, and liver dysfunction [9,10,16,17,18,19,20]. 

Preoperative diagnosis of CALI is still an unmet need [21]. Presently, the prediction of CALI relies on risk factors (e.g., chemotherapy regimen and duration) [9,10,11,16], some laboratory tests and scores (e.g., APRI score, ICG tests, and LiMax test) [10,16,19,22,23,24] and imaging modalities (e.g., heterogeneous liver parenchyma at computed tomography (CT) and magnetic resonance imaging) [25,26], but has limited accuracy. Also liver biopsy has low effectiveness because of the heterogeneous distribution of injuries and insufficient sample size [27].

In the last few years, a new approach to medical imaging has gained momentum. It is driven by the hypothesis that tissue features could be expressed on the radiological images as voxel patterns, which are invisible to the human eye. To identify these patterns, mathematical functions analyzing the spatial relation and the frequency distribution of gray levels in the voxels were developed, providing modern and specific image biomarkers [28,29]. This texture-based approach has been termed “radiomics”. However, although texture analysis has shown high accuracy in the identification of liver fibrosis [30,31,32,33], no studies have yet focused on radiomics for CALI.

The present analysis aims to investigate whether the radiomic features extracted from preoperative CT imaging can improve diagnosis of NASH and CALI in patients undergoing liver resection for CLM after preoperative chemotherapy. A defined volume of non-tumoral liver parenchyma was analyzed, thus performing a “virtual biopsy”.

## 2. Material and Methods

All consecutive patients that underwent liver resection for CLM between January 2018 and February 2020 were retrospectively considered. The following inclusion criteria were adopted: preoperative chemotherapy for at least two months; oxaliplatin- or irinotecan-based chemotherapy regimen; availability of preoperative CT for imaging review and texture analysis; preoperative imaging performed <2 months before liver resection.

The primary endpoint of the study was to analyze the contribution of radiomic analysis of liver parenchyma in the portal phase of preoperative post-chemotherapy CT scan to the diagnosis of clinically relevant CALI. Clinically relevant CALI included grade 2–3 sinusoidal dilatation, NRH of any grade, and NASH. Radiomic features were extracted from a standardized volume of interest (VOI) of non-tumoral liver parenchyma identified on the portal phase of the CT scan. The VOI was a cylinder with a basis diameter of 10 mm and a height of 25 mm outlined in the right liver between the anterior and posterior sections and positioned not to include CLM or major intrahepatic vessels. If an adequate VOI of non-tumoral parenchyma was not available in the right liver, the VOI was outlined in the left liver. In any case, a minimal distance of 20 mm between the VOI and CLM was respected. In all patients, VOI had the same shape, dimension, and requisites (no major vessels and no CLM). Radiomic features were automatically extracted from the VOI by the LifeX ® software 6 [34,35]. The study was approved by the local ethics committee and, because of its retrospective design, the need for informed consent was waived.

### 2.1. Patients Management and Pathology Data

The management of patients with CLM candidates to liver resection has been previously reported [36,37,38]. After the end of chemotherapy, patients were considered for surgery only in case of stable disease or partial response to treatment. Liver resection was performed 4 to 6 weeks after the end of chemotherapy (six weeks if bevacizumab was administered). 

In the authors’ institution, CALI are prospectively assessed in all patients undergoing resection for CLM after preoperative chemotherapy. For the present study, all specimens were reviewed by a single expert pathologist (L.d.T.). The sample of non-tumoral hepatic parenchyma was taken at a distance of at least 10 mm (20 mm whenever possible) from the tumor and the resection margin [39]. The tissue specimen (minimum area 1 cm^2^) was fixed in formalin, paraffin-embedded, and stained with hematoxylin-eosin, Masson’s trichrome, and Gomori staining. CALI evaluation was performed according to standard criteria. In details, sinusoidal dilatation was graded semi-quantitatively (from grade 0 to grade 3), according to Rubbia-Brandt et al. [8]; NRH was graded (from grade 0 to grade 3) according to the Wanless scoring system [40]; and NASH definition was based on Kleiner et al., adopting the cut-off value modified by Vauthey et al. and the scoring system of the most recent guidelines (steatohepatitis defined as the joint presence of steatosis, ballooning, and lobular inflammation ) [9,41,42] (Appendix A). Fibrosis was assessed according to the METAVIR score [43].

### 2.2. Statistical Analyses

Summary statistics were constructed with the use of frequencies and proportions for categorical data and medians and ranges for continuous variables. Chi-square (or Fisher’s exact test) and t-test (or Mann-Whitney U test) were used to evaluate potential differences in the distribution of variables according to different CALI. Univariate analysis was carried out to explore the association between the different CALI and clinical (i.e., dyslipidemia, diabetes, metabolic syndrome, body mass index (BMI), type and length of chemotherapy, age), laboratory variables (i.e., APRI, GGT), and chemotherapy details.

A multivariate logistic regression model was performed to estimate the adjusted association between each candidate predictor and the presence of different CALI (grade 2–3 sinusoidal dilatation, NRH, or steatohepatitis). Clinical rationale associated with a backward stepwise regression approach was used to retain only relevant associations. In particular: a principal component analysis (PCA) of second-order radiomic features, i.e., textural features quantifying tumor heterogeneity by analyzing the spatial distribution of pixel/voxel intensities, was performed to obtain effective predictors (Fingerprint in the following) for the model. PCA was performed on the following matrices: gray-level co-occurrence matrices (GLCM), gray-level run-length matrices (GLRLM), neighboring gray-level difference matrices (NGLDM), and gray-level zone-length matrices (GLZLM). For each of them, we retained components of the PCA that explain at least 95% of original features variability. Note that the retained fingerprints have neither clinical nor biological interpretations, but related estimates (OR and CI) may be interpreted as usual. Clinical and laboratory variables were selected according to *a priori* knowledge and the results of univariate analysis; then a stepwise regression was run, and all the variables retained by this procedure were used for predictive purposes. Continuous variables were included as continuous predictors, i.e., without assuming any categorization with arbitrary thresholds. Finally, a correlation matrix of continuous variables and correlation heat-map were generated. Correlation between features was analyzed and, whenever higher than 0.85, one of the two features involved in the correlation was removed. The final predictive model underwent internal cross-validation by splitting the series into a training set (90% of the population) and a validation set (10%). The validation procedure was repeated 100 times over 100 different samples. Results are reported in terms of mean (Std Dev) accuracy.

A decision tree was built with the variables retained by the backward stepwise selection of the multivariate model to highlight and exploit the possible nonlinear association with the outcome. Indeed, a decision tree for classification problems is a top-down greedy algorithm that divides the predictor space into distinct and non-overlapping regions (identified by the criteria/split adopted to reach each node). The slitting criteria for decision rules are defined according to the principle of minimizing the variability of the response within each node. For every observation falling into one region (node), the decision tree predicts the occurrence of the corresponding CALI of interest. Decision trees are displayed as dendrograms to highlight decision steps. Each node of the tree reports: (a) the response mode class in the node, i.e., the predicted outcome for that node (presence of CALI = 1; absence of CALI = 0, the top number in the square); (b) the percentages of observations in the node belonging to the first response class (absence of CALI) and the second response class (presence of CALI) (the two central numbers in the square, summing up to 1); (c) the percentage of the total population falling into the node (the bottom number in the square). Decision rules are specified on each split.

Stata 15 [44] and R software 1.2 [45] were used for all the analyses.

## 3. Results

In the study period (January 2018–February 2020), 78 consecutive patients that underwent liver resection for CLM after preoperative chemotherapy were enrolled in the present study. Of these, 47 (60%) patients were male, and the median age of the entire cohort was 65 (range 30–82) years. Eleven (14%) patients were obese (BMI > 30 kg/m^2^), 8 (10%) had a metabolic syndrome, and two (3%) had chronic medications potentially associated with NASH [46] (steroids in one, and tamoxifen in one). No patient had chronic medication potentially associated with NRH or sinusoidal dilatation [47,48]. CLMs were synchronous with the primary tumor in 54 (70%) patients, multiple in 70 (90%), and larger than 50 mm in 13 (17%). All patients had oxaliplatin- or irinotecan-based preoperative chemotherapy. Targeted therapies were associated with chemotherapy in 65 (83%) patients, anti-VEGF being the commonest one (58%, *n* = 45). Table 1 summarizes chemotherapy details. The median interval between chemotherapy and liver surgery was five weeks (range 4–7). All patients but four underwent minor hepatectomy. No patient had preoperative or intraoperative signs of portal hypertension. Ninety-day operative mortality rate was nil, five (6%) patients had postoperative severe morbidity and two (3%) had grade B-C liver failure.

At the final pathology examination, CALI were evident in 61 (78%) patients (Table 1). In details, grade 2–3 sinusoidal dilatation was present in 25 (32%) patients, NRH in 27 (35%), and NASH in 14 (18%). The association between CALI and patients’ characteristics, laboratory data, and chemotherapy details is reported in Table 2.

At the univariate analysis, grade 2–3 sinusoidal dilatation and NRH were associated with preoperative APRI score (median value 0.50 if grade 2–3 sinusoidal dilatation vs. 0.33 if grade 0–1, *p* = 0.006; 0.49 if NRH vs. 0.32 if not, *p* = 0.006) and were reduced in patients having received preoperative anti-VEGF therapy (32% vs. 45%, *p* = 0.030, and 24% vs. 48%, *p* = 0.027, respectively). Steatohepatitis was associated with body mass index (BMI, median value 29.9 kg/m^2^ if steatohepatitis vs. 25.4 kg/m^2^ if not, *p* < 0.001) and metabolic syndrome (50% vs. 14%, *p* = 0.013, respectively). Of the patients with chronic medications potentially associated with NASH [46], none had steatohepatitis at the final pathology evaluation. The association between postoperative outcome and CALI is reported in Appendix A. Of note, in the present series, three patients had F3 fibrosis; none had cirrhosis (F4).

The *a priori* defined characteristics of the VOI (dimension, shape, and distance from vessels and metastases) were respected in all cases. The VOIs were adequate and homogeneous in all patients.

### 3.1. Predictive Model for Sinusoidal Dilatation

At the multivariate analysis, the following clinical and laboratory variables were associated with grade 2–3 sinusoidal dilatation (Table 3): age (Odds Ratio (OR) = 1.11, Confidence Interval (CI)95% = 1.02–1.21, *p* = 0.015), anti-VEGF therapy associated with chemotherapy (OR = 0.18, CI95% = 0.04–0.77, *p* = 0.021), and APRI score (OR = 64.16, CI95% = 3.32–120.3, *p* = 0.006). In addition, three fingerprints derived from radiomic features were independent predictors of sinusoidal dilatation: GLRLM_f3 (OR = 12.25, CI95% = 1.34–111.9, *p* = 0.026), NGLDM_f1 (OR = 7.77, CI95% = 1.37–44.06, *p* = 0.021), GLZLM_f2 (OR = 0.53, CI95% = 0.31–0.91, *p* = 0.022).

The combined clinical, laboratory and radiomic model had 82% accuracy, 64% sensitivity, and 91% specificity (AUC = 0.87, Figure 1).

The model without radiomic features had AUC = 0.77 (Appendix A, delta AUC with the model including the radiomic features = −0.10). We built a decision tree based on the results of multivariate analysis (Figure 2).

The following knots were identified: age < 69 years, APRI score < 0.68, NGLDM_f2 < −0.36, Hist_IQR < 20, GLRLM_f3 ≥ 0.49. The decision tree achieved 78% accuracy, 92% sensitivity, and 72% specificity (AUC = 0.87). 

In the validation set, the multivariate logistic regression had an average accuracy of 72% (Std Dev 15%).

### 3.2. Predictive Model for NRH

At the multivariate analysis, the following clinical and laboratory variables were associated with NRH (Table 4): age (OR = 1.10, CI95% = 1.01–1.20, *p* = 0.027), BMI (OR = 0.68, CI95% = 0.49–0.94, *p* = 0.021), Irinotecan (OR = 28.71, CI95% = 1.8–459.04, *p* = 0.018), number of cycles of chemotherapy (OR = 1.15, CI95% = 1.01–1.32, *p* = 0.031), anti-VEGF therapy associated with chemotherapy (OR = 0.05, CI95% = 0.01–0.49, *p* = 0.010), and APRI score (OR = 275.08, CI95% = 4.75–15937.97, *p* = 0.007). In addition, three radiomic predictors of NRH were identified: conventional_HUQ2 (OR = 0.76, CI95% = 0.62–0.92, *p* = 0.005), GLZLM_f2 (OR = 0.05, CI95% = 0.01–0.43, *p* = 0.007), and GLZLM_f3 (OR = 7.97, CI95% = 1.52–41.85, *p* = 0.014). 

The combined clinical, laboratory and radiomic model had 85% accuracy, 81% sensitivity, and 86% specificity (AUC = 0.91, Figure 3). The model without radiomic features had AUC = 0.85 (Appendix A, delta with the model including the radiomic features = −0.06). The decision tree based on the results of multivariate logistic regression had the following knots: APRI score < 0.28, BMI ≥ 24, GLZLM_f3 < −0.3, GLZLM_f3 ≤ 0.5, GLCM_f2 < 0.094 (Figure 4). It achieved 83% accuracy, 89% sensitivity, and 80% specificity (AUC = 0.88). 

In the validation set, the multivariate logistic regression had an average accuracy of 71% (Std Dev 12%).

### 3.3. Predictive Model for NASH

At the multivariate analysis, one radiomic feature was associated with NASH (Table 5): conventional_HUQ2 (OR = 0.79, CI95% = 0.66–0.94, *p* = 0.010).

Steatohepatitis was predicted with 91% accuracy, 86% sensitivity, and 92% specificity (AUC = 0.93, Figure 5).

The model without radiomic features had AUC = 0.83 (Appendix A, delta with the model including the radiomic features = −0.10). The decision tree based on the results of multivariate analysis had a single knot, i.e., conventional_HUQ2 ≥ 99 (Figure 6). 

It achieved 86% accuracy, 93% sensitivity, and 84% specificity (AUC = 0.88). In the validation set, the multivariate logistic regression had an average accuracy of 91% (Std Dev 11%).

### 3.4. Contribution of Radiomic Features Extracted from the Unenhanced CT Scan

In the same setting of 78 patients, we analyzed the performances of models considering radiomic features extracted from the unenhanced CT scans. No textural parameter was predictive of grade 2–3 sinusoidal dilatation. Considering NRH, the following independent predictors were identified: Hist_IQR (OR 0.55, CI = 0–34–0.89, *p* = 0.01), NGLDM_f2 (OR 0.16, CI = 0.04–0.067 *p* = 0.012), and GLZLM_f5 (OR 30.46 CI = 1.24–745.57 *p* = 0.036). The model identified at the multivariate analysis had 82% accuracy, 74% sensitivity, and 86% specificity (AUC = 0.91). In the validation set, the model had an average accuracy of 71% (Std Dev 12%). Considering NASH, conventional HUQ2 (OR 0.52, CI = 0.31–0.87 *p* = 0.009) was confirmed as independent predictor, with a 94% accuracy, 93% sensitivity and 94% specificity (AUC = 0.99). In the validation set, the model had an average accuracy of 85% (Std Dev 10%).

## 4. Discussion

An accurate non-invasive diagnosis of NASH and CALI is a relevant unmet need for clinicians. Standard imaging modalities provide a reliable diagnosis of steatosis, but not of NASH and CALI [25,26]. Some radiological signs of sinusoidal dilatation and NRH have been depicted, such as liver parenchyma heterogeneity and focal hyperintensities at magnetic resonance imaging, liver atrophy after chemotherapy at CT scan, and splenomegaly, but their assessment is not standardized [25,49,50,51,52]. Also a percutaneous liver biopsy has low reliability in identifying liver injuries other than steatosis because of their heterogeneous distribution and the small amount of sampled tissue available [27]. To date, CALI prediction relies on patients’ history, i.e., chemotherapy regimen and the number of administered cycles, and on some liver function tests, such as APRI score or ICG test, even if the results of these tools are misleading in up to one-third of patients [10,11,16,19,22,23,24].

Recently, image mining and analysis have presented new perspectives. During the past decades, several approaches - statistical, geometrical, structural, and model-based methods to transform-based techniques - have been explored to extract quantitative information from images and develop potential non-invasive biomarkers to detect and characterize diseases [53]. Above all, the use of grey level co-occurrence and higher-order matrices has rooted in clinical research as texture descriptors, although their adoption has generally turned into automatic feature extraction tools, namely radiomics [54]. Radiomics involve the definition of mathematical features able to capture data on the grey-scale patterns, interpixel relationships, shape, and spectral properties within regions of interest on radiological images [55]. This technique allows researchers to access standardized texture information about images and to carry out informed inference, aiding traditional clinical investigations. To date, radiomics effectively predict biological characteristics and outcomes of several diseases [28,56,57]. In CLM patients, evidence is preliminary, but texture analysis not only improves prediction of survival compared to standard prognosticators but also provides earlier and more accurate prediction of response to chemotherapy than RECIST criteria [29]. Radiomics also accurately identifies fibrosis when applied to liver parenchyma analysis [30,31,32,33]. However, no study analyzed the association of textural features with CALI.

Radiomics are expected to detect CALI-related tissue heterogeneity and alterations, as confirmed by the present analysis. Considering sinusoidal dilatation and NRH, radiomic signatures improved the diagnosis of liver injuries achieved by standard clinical and laboratory parameters: the inclusion of radiomic predictors in the multivariate model increased the AUC by 0.10 for sinusoidal dilatation (overall AUC 0.87) and by 0.06 for NRH (AUC 0.91). Soubrane et al. reported similar performances of the APRI score for the prediction of sinusoidal dilatation (AUC 0.85) [19], but such good results were not confirmed by other series. The present study demonstrated that radiomic features integrate the “traditional” predictors, i.e., APRI score, anti-VEGF therapies, and chemotherapy duration [10,12,16,19]. The performances of the combined model were promising for sinusoidal dilatation (64% sensitivity, 82% accuracy), and very good for NRH (81% and 85%, respectively). The adoption of a decision tree further improved the diagnosis of sinusoidal dilatation (92% sensitivity), better exploiting potential nonlinear associations of the radiomic predictors with the outcome. Split-oriented decision steps in tree-based models not only optimized results but also provided an easy-to-handle tool that substantiates the clinical applicability of radiomics. Radiomic features associated with sinusoidal dilatation and NRH (derived from NGLDM, GLRLM, GLZLM, and GLCM matrices) catch the tissue heterogeneity, expressed as grey-level variability between one voxel and its neighbors in the three-dimensions and the homogeneity of runs of voxels in two or three dimensions. This is in line with pathology data that depict irregular sinusoidal congestion and nodular area, and with data of magnetic resonance imaging that show some irregular parenchymal enhancement [8,49].

Significant data were also obtained for NASH. The Hounsfield Q2 values (HUQ2, i.e., the Hounsfield value of the second quartile, median value) led to a highly reliable diagnosis of steatohepatitis (86% sensitivity, 92% specificity). The addition of radiomic features to standard clinical predictors increased the AUC by 0.10 with the identified cut-off value associated with extremely high performances (93% sensitivity, 86% accuracy). HUQ2 measures the intensity of tissue signal within a certain attenuation range that is coherent with the presence of steatosis, the most relevant component of NASH [26]. This parameter is more a statistical measure rather than a higher order textural feature. Nonetheless, the present analysis provided a cut-off value of HUQ2 that made it a usable tool for NASH diagnosis. We obtained the same results and performances (HUQ2, accuracy 94%) when the textural features extracted from unenhanced CT scans were considered, suggesting that radiomics may adequately capture these tissue characteristics even without the need for contrast enhancement. 

The present study also proposed a virtual biopsy of the non-tumoral liver, an easy-to-collect standardized VOI with a large tissue sample (25 × 10 mm, approximatively 2 cm^3^ of liver tissue), much larger than actual liver biopsies and enough to catch the heterogeneous distribution of CALI. It is true that a radiomic analysis of the entire liver could be more exhaustive but this would require a much more complex segmentation with a time-consuming exclusion of intrahepatic vessels and tumors and significantly greater computational power. In the future, the implementation of AI-driven segmentation protocols and the advent of quantum computing will probably overcome these limitations. The present virtual biopsy is highly reproducible and the adoption of software with automatic extraction of radiomic features (LifeX^®^) increases the potential diffusion of this approach, even if the interpretability and explainability of radiomic data are still debated.

A reliable assessment of NASH and CALI could have a consistent impact on clinical practice. Sinusoidal dilatation, NRH, and steatohepatitis are associated with an increased risk of intraoperative bleeding, postoperative liver failure, and mortality. The surgical strategy could be adapted to the characteristics of the non-tumoral liver [21]. Higher cut-off volume values of the future liver remnant could be pursued in the presence of CALI or NASH, leading to extended indications to preoperative portal vein occlusion [58]. The presence of portal hypertension should be excluded in patients with NASH and NRH [10,15,59]. Further clinical applications of the proposed virtual biopsy can be anticipated. CALI have some correlations with the effectiveness of chemotherapy (the higher the grade of sinusoidal injury, the lower the pathological response rate) and their identification is essential to further understand this association [11,60]. An accurate diagnosis of CALI is needed in case of CLM recurrence to evaluate the liver tolerance to new chemotherapy lines and repeat surgery [61]. Steatosis and NASH have an increasing incidence in Western countries because of their close relationship with obesity and metabolic syndrome and may lead to liver dysfunction and tumors in the long term [14,15,62]. The virtual biopsy could be helpful to monitor liver injuries and identify patients at risk for complications. Similarly, some chronic medications may lead to steatohepatitis [46]. Steatosis and NASH have shown some reversibility with lifestyle modifications, correction of metabolic disorders, and bariatric surgery [14,63,64], but non-invasive diagnosis of CALI is mandatory to verify the effectiveness of any treatment.

The present analysis has some limitations to address. First, it is an exploratory retrospective study with a limited number of patients. Data are preliminary and require an external validation but it is important to note that they are based on some robust foundations: CALI had a standardized and prospective evaluation, patients were treated over a short two-year period with homogeneous schedules, standard predictors of CALI were confirmed together with the new contribution of radiomic signatures, and internal validation provided encouraging confirmation of good performances. A second limitation is that some predictors of CALI were not tested, such as the ICG test or LiMax test, but they are not standard in clinical practice and are expected to give a contribution similar to the APRI score. A third is related to the reproducibility of data that could be reduced by the heterogeneity of imaging techniques among institutions, even if the CT scan is the most standardized imaging modality. Finally, the usability of radiomic features remains an issue. Our explorative analysis provided intriguing data, but we are still far from accomplishing a real clinical application of radiomics, which remains the challenge of research in the near future. Radiomics does indeed suffer from closed-source nature, unharmonized acquisition settings, discordant reconstruction parameters, lack of interpretability, redundancy, and methodological bias [54,65,66]. A wide and active research area is growing around grey-level quantization and pre-processing, aiming at informative rather than descriptive statistics from images. Such studies could open new perspectives in clinical applications of medical imaging analysis.

## 5. Conclusions

In conclusion, it was observed that a radiomic signature based on the texture analysis of liver parenchyma might improve diagnosis of sinusoidal dilatation, NRH, and steatohepatitis. Although the application of radiomics to clinical practice is still to be accomplished, our preliminary data can provide a basis for an innovative precision medicine approach to patients at risk for liver injuries.

## Figures and Tables

**Figure 1 cancers-13-03077-f001:**
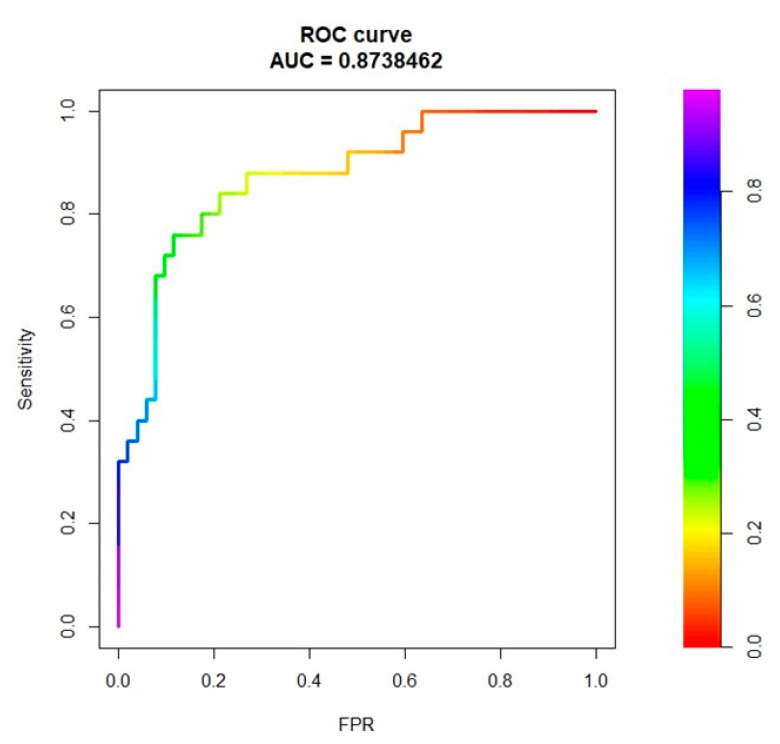
ROC curve analysis referring to the prediction of sinusoidal dilatation (grade 0–1 vs. grade 2–3), considering clinical, laboratory, and radiomic variables for model training.

**Figure 2 cancers-13-03077-f002:**
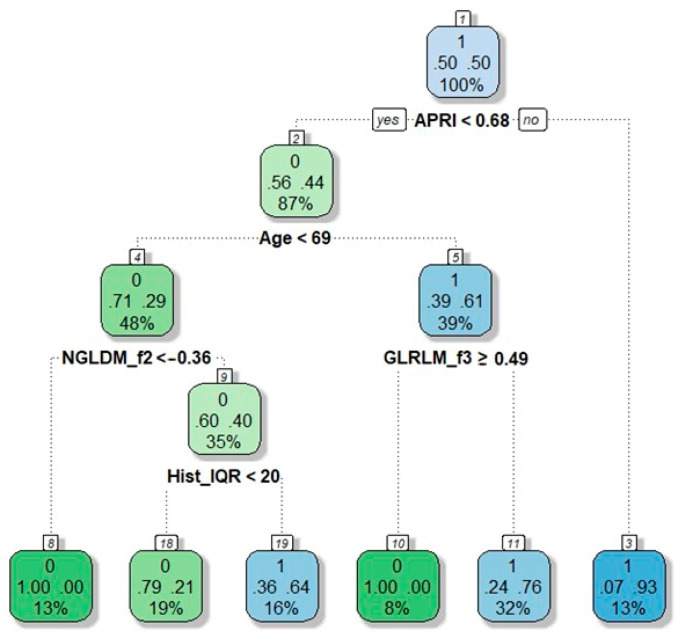
Decision tree for the prediction of grade 0–1 vs. grade 2–3 sinusoidal dilatation (based on model in Table 3). Nodes correspond to the decision steps. Each colored square reports: (a) the response mode class in the node, i.e., the predicted outcome of that node (presence of grade 2–3 sinusoidal dilatation = 1; absence of grade 2–3 sinusoidal dilatation = 0, the top number in the square); (b) the percentages of observations in the node belonging to the first response class (absence of grade 2–3 sinusoidal dilatation) and the second response class (presence of grade 2–3 sinusoidal dilatation) (the two central numbers in the square, summing up to 1); (c) the percentage of the total population falling into the node (the bottom number in the square). Decision rules are specified on each node.

**Figure 3 cancers-13-03077-f003:**
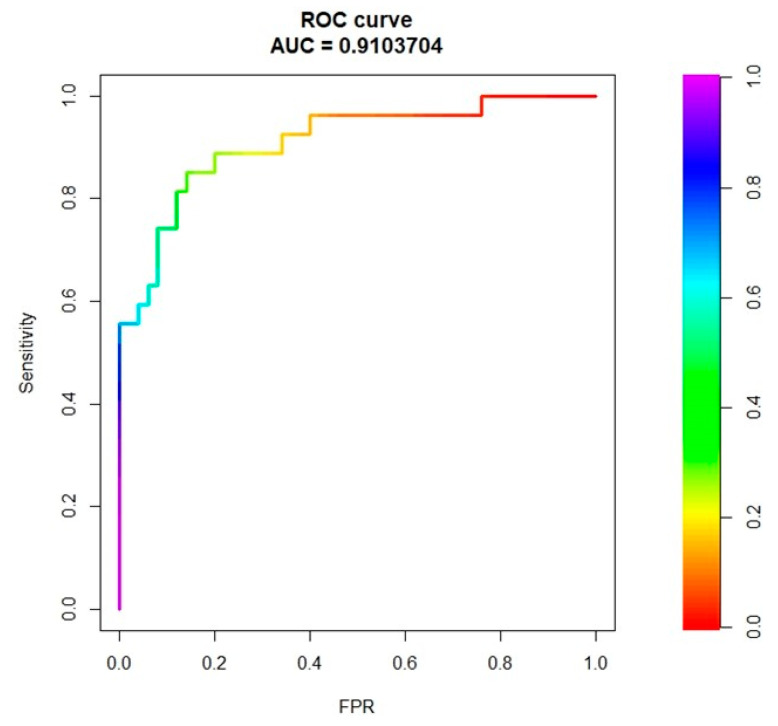
ROC curve analysis referring to the prediction of NRH (no vs. yes), considering clinical, laboratory, and radiomic variables for model training.

**Figure 4 cancers-13-03077-f004:**
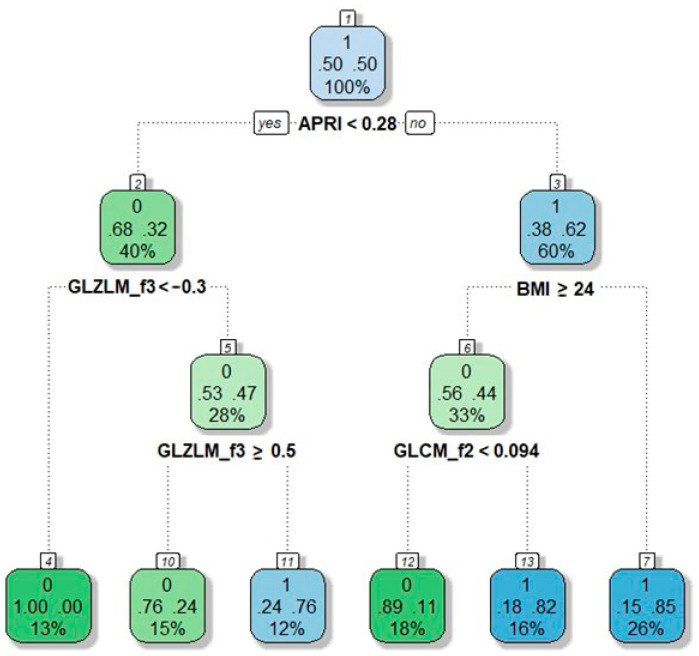
Decision tree for the prediction of NRH (based on model in Table 4). Nodes correspond to the decision steps. Each colored square reports: (a) the response mode class in the node, i.e., the predicted outcome of that node (presence of NRH =1; absence of NRH =0, the top number in the square); (b) the percentages of observations in the node belonging to the first response class (absence of NRH) and the second response class (presence of NRH) (the two central numbers in the square, summing up to 1); (c) the percentage of the total population falling into the node (the bottom number in the square). Decision rules are specified on each node.

**Figure 5 cancers-13-03077-f005:**
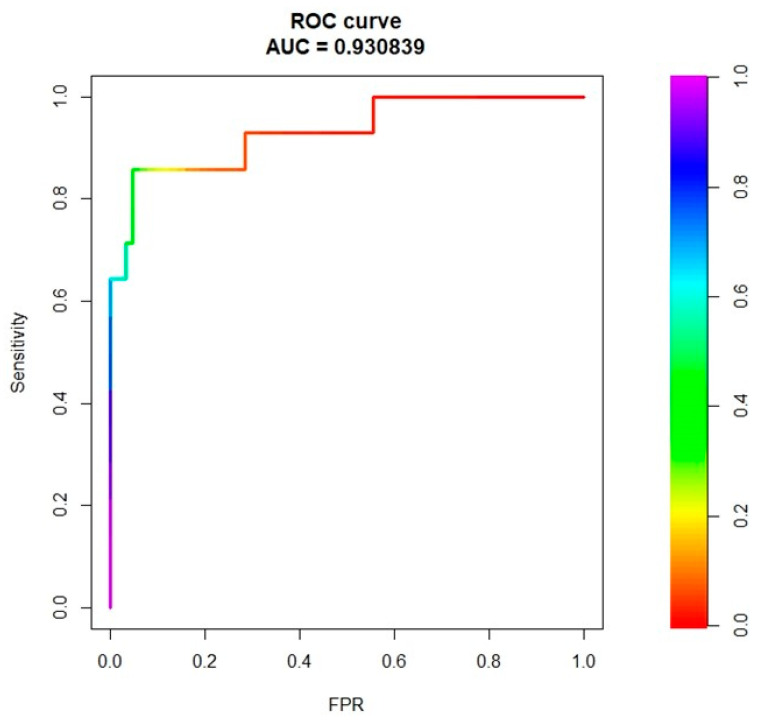
ROC curve analysis referring to the prediction of NASH (yes vs. no), considering clinical, laboratory, and radiomic variables for model training.

**Figure 6 cancers-13-03077-f006:**
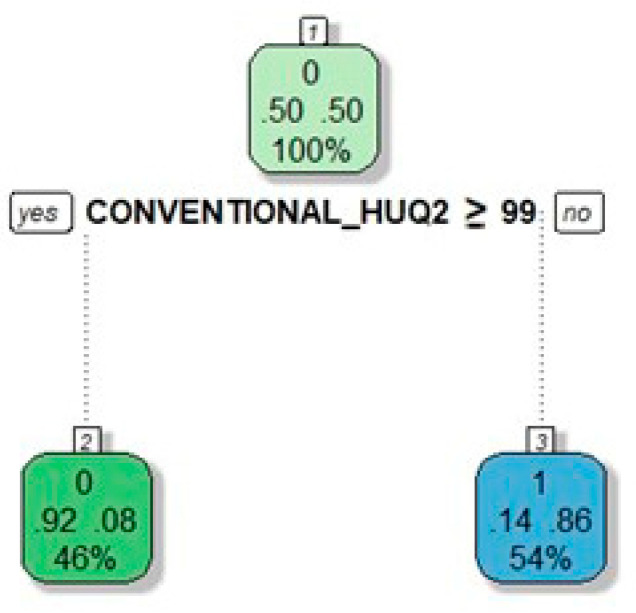
Decision tree for the prediction of NASH (based on model in Table 5). Nodes correspond to the decision steps. Each colored square reports: (a) the response mode class in the node, i.e., the predicted outcome of that node (presence of NASH =1; absence of NASH =0, the top number in the square); (b) the percentages of observations in the node belonging to the first response class (absence of NASH) and the second response class (presence of NASH) (the two central numbers in the square, summing up to 1); (c) the percentage of the total population falling into the node (the bottom number in the square). Decision rules are specified on each node.

**Table 1 cancers-13-03077-t001:** Details of chemotherapy and pathology data.

Chemotherapy Data	
**Regimen**	
Oxaliplatin	69 (88%)
Irinotecan	33 (42%)
Anti-VEGF treatment	45 (58%)
Anti-EGFR treatment	29 (37%)
Number of cycles, median (range)	8 (3–35)
>6 cycles	48 (62%)
≥2 lines	25 (32%)
Interval chemotherapy-surgery, weeks, median (range)	5 (4–7)
**CALI**	
Sinusoidal dilatation	56 (72%)
Grade 2–3 *	25 (32%)
NRH	27 (35%)
Grade 2–3 **	9 (12%)
Steatosis	34 (44%)
Grade 2–3 ***	19 (24%)
Lobular inflammation	21 (27%)
Hepatocellular ballooning	14 (18%)
NASH ****	14 (18%)

VEGF, Vascular Endothelial Growth Factor; EGFR, Epidermal Growth Factor Receptor; CALI, Chemotherapy-Associated Liver Injury; NRH, Nodular Regenerative Hyperplasia; NASH, Non-alcoholic Steatohepatitis; * according to Rubbia-Brandt et al. [8]; ** according to the Wanless scoring system [40]; *** according to Kleiner et al. [41]; **** according to EASL guidelines, Marchesini et al. [42].

**Table 2 cancers-13-03077-t002:** Association between CALI and clinical and laboratory data.

**Grade 2–3 Sinusoidal Dilatation**				
		**N**	**Y**	***p***
Age, years, median (range)		61 (30–82)	66 (51–80)	0.032
APRI score, median (range)		0.33 (0.10–1.16)	0.50 (0.12–1.89)	0.006
GGT, UI/L, median (range)		59 (7–247)	76 (11–372)	0.290
BMI, kg/m^2^, median (range)		25.8 (19.0–40.3)	24.9 (20.4–33.6)	0.368
Dyslipidemia	N	24 (69%)	11 (31%)	0.915
	Y	29 (67%)	14 (33%)	
Diabetes	N	44 (66%)	23 (34%)	0.487
	Y	9 (82%)	2 (18%)	
Metabolic syndrome	N	47 (67%)	23 (33%)	0.652
	Y	6 (75%)	2 (25%)	
Oxaliplatin-based chemotherapy	N	7 (78%)	2 (22%)	0.502
	Y	46 (67%)	23 (33%)	
Irinotecan-based chemotherapy	N	30 (67%)	15 (33%)	0.777
	Y	23 (70%)	10 (30%)	
Anti-VEGF treatment	N	18 (55%)	15 (45%)	0.030
	Y	35 (78%)	25 (32%)	
Number of cycles of chemotherapy	1–6	19 (63%)	11 (37%)	0.490
	>6	34 (71%)	14 (29%)	
**NRH**				
		**N**	**Y**	***p***
Age, years, median (range)		62 (30–82)	64 (47–80)	0.333
APRI score, median (range)		0.32 (0.10–1.16)	0.49 (0.12–1.89)	0.006
GGT, UI/L, median (range)		54 (7–247)	83 (11–372)	0.032
BMI, kg/m^2^, median (range)		26.0 (19.0–40.3)	4.6 (20.4–32.3)	0.161
Dyslipidemia	N	21 (60%)	14 (40%)	0.367
	Y	30 (70%)	13 (30%)	
Diabetes	N	43 (64%)	24 (36%)	0.739
	Y	8 (73%)	3 (27%)	
Metabolic syndrome	N	44 (63%)	26 (37%)	0.165
	Y	7 (87%)	1 (13%)	
Oxaliplatin-based chemotherapy	N	7 (78%)	2 (22%)	0.406
	Y	44 (64%)	25 (36%)	
Irinotecan-based chemotherapy	N	31 (69%)	14 (31%)	0.447
	Y	20 (61%)	13 (39%)	
Anti-VEGF treatment	N	17 (52%)	16 (48%)	0.027
	Y	34 (76%)	11 (24%)	
Number of cycles of chemotherapy	1–6	21 (70)	9 (30%)	0.498
	>6	30 (62%)	18 (38%)	
**Steatohepatitis**				
		**N**	**Y**	***p***
Age, years, median (range)		63 (30–82)	61 (47–78)	0.595
APRI score, median (range)		0.37 (0.10–1.89)	0.42 (0.14–1.16)	0.610
GGT, UI/L, median (range)		63 (7–372)	72 (21–218)	0.651
BMI, kg/m^2^, median (range)		25.4 (19.0–33.7)	29.9 (22.6–40.3)	<0.001
Dyslipidemia	N	31 (89%)	4 (11%)	0.239
	Y	33 (77%)	10 (23%)	
Diabetes	N	55 (82%)	12 (18%)	1.000
	Y	9 (82%)	2 (18%)	
Metabolic syndrome	N	60 (86%)	10 (14%)	0.013
	Y	4 (50%)	4 (50%)	
Oxaliplatin-based chemotherapy	N	7 (78%)	2 (22%)	0.722
	Y	57 (83%)	12 (17%)	
Irinotecan-based chemotherapy	N	40 (89%)	5 (11%)	0.066
	Y	24 (73%)	9 (27%)	
Anti-VEGF treatment	N	31 (94%)	2 (6%)	0.019
	Y	33 (73%)	12 (27%)	
Number of cycles of chemotherapy	1–6	25 (83%)	5 (17%)	0.816
	>6	39 (81%)	9 (19%)	

APRI, AST-to-Platelet Ratio Index; GGT, Gamma-glutamyltransferase; BMI, Body Mass Index; VEGF, Vascular Endotelial Growth Factor; NRH, Nodular Regenerative Hyperplasia; N refers to NO; Y refers to YES.

**Table 3 cancers-13-03077-t003:** Multivariate analysis of predictors of grade 2–3 sinusoidal dilatation.

Variable.	OR (95% IC)	*p*
Age	1.11 (1.02–1.21)	0.015
APRI score	64.16 (3.32–120.30)	0.006
Oxaliplatin-based chemotherapy	11.92 (0.54–26.29)	0.118
Irinotecan-based chemotherapy	3.46 (0.66–18.18)	0.142
Anti-VEGF treatment	0.18 (0.04–0.77)	0.021
Number of cycles of chemotherapy	1.08 (0.98–1.2)	0.128
Hist_IQR	0.74 (0.49–1.11)	0.144
GLRLM_f3	12.25 (1.34–111.90)	0.026
NGLDM_f1	7.77 (1.37–44.06)	0.021
NGLDM_f2	0.28 (0.04–1.73)	0.169
GLZLM_f2	0.53 (0.31–0.91)	0.022
GLZLM_f4	1.72 (0.85–3.48)	0.131

APRI, AST-to-Platelet Ratio Index; VEGF, Vascular Endothelial Growth Factor; Hist_IQR, Histogram Interquartile Range; GLRLM_f3, Grey-Level Run Length Matrix Fingerprint 3; NGLDM_f1, Neighborhood Grey-Level Different Matrix Fingerprint 1; NGLDM_f2, Neighborhood Grey-Level Different Matrix Fingerprint 2; GLZLM_f2, Grey-Level Zone Length Matrix Fingerprint 2; GLZLM_f4, Grey-Level Zone Length Matrix Fingerprint 4.

**Table 4 cancers-13-03077-t004:** Multivariate analysis of predictors of NRH.

Variable	OR (95% IC)	*p*
Age	1.10 (1.01–1.20)	0.027
APRI score	275.08 (4.75–15937.97)	0.007
BMI	0.68 (0.49–0.94)	0.021
Oxaliplatin-based chemotherapy	34.41 (0.52–2295.05)	0.099
Irinotecan-based chemotherapy	28.71 (1.80–459.04)	0.018
Anti-VEGF treatment	0.05 (0.01–0.49)	0.010
Number of cycles of chemotherapy	1.15 (1.01–1.32)	0.031
CONVENTIONAL_HUQ2	0.76 (0.62–0.92)	0.005
GLCM_f2	1.99 (0.84–4.71)	0.119
GLRLM_f3	0.39 (0.11–1.42)	0.153
NGLDM_f2	2.65 (0.86–8.24)	0.091
GLZLM_f2	0.05 (0.01–0.43)	0.007
GLZLM_f3	7.97 (1.52–41.85)	0.014

APRI, AST-to-Platelet Ratio Index; BMI, Body Mass Index; VEGF, Vascular Endothelial Growth Factor; HUQ2, Hounsfield Unit Quartile 2; GLCM_f2, Gray - Level Co-occurrence Matrix Fingerprint 2; GLRLM_f3, Grey-Level Run Length Matrix Fingerprint 3; NGLDM_f2, Neighborhood Grey-Level Different Matrix Fingerprint 2; GLZLM_f2, Grey-Level Zone Length Matrix Fingerprint 2; GLZLM_f3, Grey-Level Zone Length Matrix Fingerprint 3.

**Table 5 cancers-13-03077-t005:** Multivariate analysis of predictors of NASH.

Variable	OR (95% IC)	*p*
CONVENTIONAL_HUQ2	0.79 (0.66–0.94)	0.010
GLZLM_f2	0.22 (0.03–1.66)	0.143

HUQ2, Hounsfield Unit Quartile 2; GLZLM_f2, Grey-Level Zone Length Matrix Fingerprint 2.

## Data Availability

The data presented in this study are available on request from the corresponding author. The data are not publicly available due to privacy restrictions.

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
