# Peer review of "Virtual Biopsy for Diagnosis of Chemotherapy-Associated Liver Injuries and Steatohepatitis: A Combined Radiomic and Clinical Model in Patients with Colorectal Liver Metastases"

_cancers, 2021, doi:10.3390/cancers13123077_

Round 1
Reviewer 1 Report
This is an interesting manuscript, focusing on the predictive potential of radiomic parameters for chemotherapy associated liver injury. While the concept seems appealing there are multiple critical issues in this manuscript that very much limits the relevance of these observations:
Major:
Our major issues with this manuscript is the sample size of only 78 patients –the dividing the cohort in 3 different types of CALI and then even trying to validate results in as I understand 10% of the patients. This is simply insufficient.
Otherwise the entire analyses of this paper are prone to overfitting – as all exploratory analyses of markers are. It appears almost sketchy that the authors further develop a decision tree that then further multiplies the issue of overfitting – without EXTERNAL validation – these observations have a high risk to strikingly overstate the predictive potential of these parameters.
Given the descriptive nature the authors should add a direct comparison of the AUCs of their clinical parameters with their radiomic parameters.
The manuscript puts postoperative outcome after liver resection in center for the clinical relevance of the reported results. However, no postoperative outcome associations are reported. As this represents the ultimate clinical relevance of these finding these associations should be shown.
Radiologic imaging techniques can strikingly vary between institutions – this further stresses the fact that external validation is critical – given the retrospective nature for the analyses this should be relatively easily possible.
Minor:
Multiple figures refer to multiple numbers but that are not sufficiently explained. The figure legends need to significantly improve.
Author Response
“Virtual biopsy for diagnosis of chemotherapy-associated liver injuries and steatohepatitis. A combined radiomic and clinical model in patients with colorectal liver metastases.”
Point-by-point response to reviewer #1
Review Report in bold
Author's Notes to Reviewer in italic
This is an interesting manuscript, focusing on the predictive potential of radiomic parameters for chemotherapy associated liver injury. While the concept seems appealing there are multiple critical issues in this manuscript that very much limits the relevance of these observations:
Major:
Our major issues with this manuscript is the sample size of only 78 patients – the dividing the cohort in 3 different types of CALI and then even trying to validate results in as I understand 10% of the patients. This is simply insufficient.
We thank the reviewer for raising the issue, which let us to better clarify the experimental setting. We set up a different model for each of the three CALI, but in all these cases we made use of the entire dataset. Therefore, the dataset is NOT divided into sub-cohorts, and the validation procedure (as better clarified in the following) is performed at the best of the dataset potential, i.e., through an internal (multi-fold) cross-validation.
We are perfectly aware that the number of observations represents one of the main limitations of this work (as stressed in lines 310-313), but we firstly aim at producing preliminary evidence in a field (radiomics applied to chemotherapy-induced damages) that is definitively requiring proofs of concept.
Otherwise the entire analyses of this paper are prone to overfitting – as all exploratory analyses of markers are. It appears almost sketchy that the authors further develop a decision tree that then further multiplies the issue of overfitting – without EXTERNAL validation – these observations have a high risk to strikingly overstate the predictive potential of these parameters.
As mentioned above, unfortunately, we do not have an external dataset to carry out the required external validation. Nevertheless, the internal cross-validation performed for the multivariate logistic analysis is exactly aimed at reducing the risk of overfitting, being the best proxy for the external misclassification error that may be obtained from an internal procedure.
The validation procedure was repeated 100 times over 100 different samples, obtained by splitting the series into a training set (90% of the population) and a validation set (10%). Results of this validation procedure are reported in terms of mean and std. dev. accuracy for each model, in the corresponding part of the results.
Given the descriptive nature the authors should add a direct comparison of the AUCs of their clinical parameters with their radiomic parameters.
We thank the reviewer for this suggestion. One of the main aims of the paper is exactly to provide evidence for the improvement of CALI prediction by combining radiomics with standard clinical parameters. In the revised version of the paper, we provided the comparison between the AUCs of the models with radiomics+clinical covariates vs the AUCs of the models with clinical covariates only. In particular, the Delta AUCs of the two models (radiomics+clinical vs. clinical-only) for each CALI are provided. We reported them in the results paragraph of the abstract and, more in detail, in Sections 2.1 (line 140), 2.2 (line 169), and 2.3 (197) of the Results for the three CALI, and Figures S1, S2, and S3. Results show that the inclusion of radiomic signatures in the multivariate model increases the AUC of 0.10 for the sinusoidal dilatation, 0.06 for NRH, and 0.10 for Steatohepatitis. We also mentioned these data in the discussion (lines 254-256 and 274).
The manuscript puts postoperative outcome after liver resection in center for the clinical relevance of the reported results. However, no postoperative outcome associations are reported. As this represents the ultimate clinical relevance of these finding these associations should be shown.
The association of the postoperative outcome with CALI is well established in the literature and is the most clinically relevant aspect of liver injuries. We had the chance to collect a series of patients with very few postoperative events: no (0/78 patients) 90-day mortality, 6% (5/78) severe morbidity, and 3% (2/78) liver failure. Accordingly, we decided not to analyze the association between CALI and outcome in the original manuscript. As suggested by the reviewer, we added this analysis in the revised version. We mentioned it in the manuscript (results lines 104-105 and lines 118), and we provided a supplementary table (table S2) reporting outcomes and exploring their associations with CALI. A trend of increased liver failure rate was observed in patients with CALI in comparison with patients without CALI, even if it did not reach statistical significance because of the small number of events. We want to stress again that our study aimed to provide a preliminary assessment of the power of radiomics to preoperatively identify CALI, whose association with the outcomes has been extensively proven yet.
Radiologic imaging techniques can strikingly vary between institutions – this further stresses the fact that external validation is critical – given the retrospective nature for the analyses this should be relatively easily possible.
We thank the reviewer for this comment. We agree that radiologic imaging techniques may vary between institutions, even if CT scan is the most standardized imaging modality. For sure, external validation is needed to confirm the reproducibility of our preliminary data. We added a commentary about this topic in the discussion (see lines 314-315).
Minor:
Multiple figures refer to multiple numbers but that are not sufficiently explained. The figure legends need to significantly improve.
In the revised version of the paper, we improved the legends of the Figures providing a wider amount of information, thus making them more self-explaining.
Reviewer 2 Report
Dear Editor,
I read with great interest the paper entitled: “Virtual Biopsy for Diagnosis of Chemotherapy-Associated 2 Liver Injuries and Steatohepatitis. A Combined Radiomic and 3 Clinical Model in Patients with Colorectal Liver Metastases”.
The discovery of non-invasive tools in the diagnosis of liver disease is a topic of great importance and in the oncologic setting the possibility to promptly identify liver impairment could help fine tuning the treatment.
There are, however, some major comments.
- In this retrospective study, resection specimens were used. A consecutive liver biopsy of the non-tumoral liver parenchyma is, of course, not available due to the retrospective design of the study. There are some aspect that need to be considered. Firstly, it is not mentioned in the paper which is the distance between the resection margin and the analyzed non-tumoral tissue. Secondly, the authors do not mention the length and the number of portal tracts of the histologic non-tumoral specimen analysed. This aspect should be clarified since it is essential in the interpretation of the liver histology. Moreover, in the specific setting of NRH, the morphological changes may not be striking in needle core biopsies (Bakshi et al, Pathol Res Pract 2020) and therefore the characteristics of the histologic specimen are very important.
- The authors analyse also the presence of NASH. From a histologic point of view they define NASH as NAS score of 4 or more. However, the diagnosis of NASH requires the joint presence of steatosis, ballooning and lobular inflammation as reported the the latest guidelines (EASL guidelines, J Hepatol 2016; Kleinier et al, Hepatology 2005; Kleiner et al, Semin Liver Dis 2012; Bedossa et al, Hepatology 2014). It would be better to use the latter definition. Moreoever fibrosis has not been reported in the histology analysis, this should be at least described.
- Furthermore, it would be important to report the presence of Non-alcoholic fatty liver disease (NAFLD) in the medical history before chemotherapy to make a distinction between Non-alcoholic steatohepatitis (NASH) and CASH (chemotherapy associated steatohepatitis). The authors report the presence of the metabolic syndrome, but it is also of note that some patients are obese and in some cases morbid obese (and might have a priori NAFLD/NASH). Is this the case?
- The authors should report the use of drugs other than chemotherapy that could have influenced the histology picture.
- The authors conclude that CALI is associated with increased risk of intraoperative bleeding and of postoperative liver failure and mortality. The presence of fibrosis F3-4 should be reported and data on portal hypertension should be added if available. It is known that there is an increase of portal pressure in NASH patients also in non-cirrhotic patients.
Minor points:
- Please explain in extenso the abbreviations all over the paper (e.g. VOI, SinDil…)
- I suggest to change “lobular flogosis” with “lobular inflammation” (of course “flogosis” is an elegant term and it has the same meaning but “inflammation” is the term commonly used in the definitions and guidelines and it would be better to align the terminology to the literature).
- Table 1: it is visually not very clear what “grade 2-3” refers to. Please consider to reformulate.
Author Response
“Virtual biopsy for diagnosis of chemotherapy-associated liver injuries and steatohepatitis. A combined radiomic and clinical model in patients with colorectal liver metastases.”
Point-by-point response to reviewer #2
Review Report in bold
Author's Notes to Reviewer in italic
Dear Editor,
I read with great interest the paper entitled: “Virtual Biopsy for Diagnosis of Chemotherapy-Associated 2 Liver Injuries and Steatohepatitis. A Combined Radiomic and 3 Clinical Model in Patients with Colorectal Liver Metastases”.
The discovery of non-invasive tools in the diagnosis of liver disease is a topic of great importance and in the oncologic setting the possibility to promptly identify liver impairment could help fine tuning the treatment.
There are, however, some major comments.
- In this retrospective study, resection specimens were used. A consecutive liver biopsy of the non-tumoral liver parenchyma is, of course, not available due to the retrospective design of the study. There are some aspects that need to be considered. Firstly, it is not mentioned in the paper which is the distance between the resection margin and the analyzed non-tumoral tissue. Secondly, the authors do not mention the length and the number of portal tracts of the histologic non-tumoral specimen analysed. This aspect should be clarified since it is essential in the interpretation of the liver histology. Moreover, in the specific setting of NRH, the morphological changes may not be striking in needle core biopsies (Bakshi et al, Pathol Res Pract 2020) and therefore the characteristics of the histologic specimen are very important.
We thank the reviewer for his/her comment. As highlighted by the reviewer, the assessment of CALI was performed on the resected specimen, and not on a liver biopsy. We analyzed a large sample of non-tumoral tissue, at least 1 cm2. The sample was taken at a distance of at least 10 mm (20 mm whenever possible) from the tumor and the resection margin. These data are better detailed in Methods (lines 351-353).
- The authors analyse also the presence of NASH. From a histologic point of view they define NASH as NAS score of 4 or more. However, the diagnosis of NASH requires the joint presence of steatosis, ballooning and lobular inflammation as reported the the latest guidelines (EASL guidelines, J Hepatol 2016; Kleinier et al, Hepatology 2005; Kleiner et al, Semin Liver Dis 2012; Bedossa et al, Hepatology 2014). It would be better to use the latter definition.
As suggested, we adopted the definition of NASH reported in the latest guidelines. After the re-classification, the diagnosis of NASH remained the same. Accordingly, no modification of the results occurred. We modified the definition of NASH in Methods (see Line 356-358).
- Moreover fibrosis has not been reported in the histology analysis, this should be at least described.
As suggested, we reported the assessment of fibrosis. According to the following comment, we adopted the METAVIR score. This has been detailed in Methods (Line 358). Three patients had fibrosis F3, none had cirrhosis (F4). This has been mentioned in the Results (line 118-119).
- Furthermore, it would be important to report the presence of Non-alcoholic fatty liver disease (NAFLD) in the medical history before chemotherapy to make a distinction between Non-alcoholic steatohepatitis (NASH) and CASH (chemotherapy associated steatohepatitis). The authors report the presence of the metabolic syndrome, but it is also of note that some patients are obese and in some cases morbid obese (and might have a priori NAFLD/NASH). Is this the case?
We thank the reviewer for this comment. It is an important point, and we completely agree that steatohepatitis can be due to risk factors other than chemotherapy. However, we analyzed patients affected by colorectal liver metastases and no patient had a biopsy of the non-tumoral liver parenchyma before chemotherapy. The assessment of NAFLD before chemotherapy is not available as well. The distinction between NASH and CASH in this setting is almost impossible. We can just analyze, as we did (line 96-98 and 115-116, table S1), the co-existing risk factors, such as obesity or metabolic syndrome. This should not be considered a major limitation of the present analysis. First, our study aimed to elucidate the contribution of radiomics to the identification of liver injuries; their detection and clinical significance are independent of etiology (CASH or NASH). Second, we included in the univariate analysis (lines 115-116, Table S1) and in the multivariate one also the metabolic parameters, exactly to assess their impact.
- The authors should report the use of drugs other than chemotherapy that could have influenced the histology picture.
We added data on chronic medications that could have determined liver injuries. We considered drugs that could induce NASH, NRH, or sinusoidal injuries, according to the LiverTox NIH database. Two (3%) patients had chronic medications potentially associated with NASH (steroids in one, and tamoxifen in one). None of these patients had NASH at the final pathology evaluation. No patient had chronic medications potentially associated with sinusoidal dilatation or NRH. These data were added in the Results (lines 97-99 and 116-117).
- The authors conclude that CALI is associated with increased risk of intraoperative bleeding and of postoperative liver failure and mortality. The presence of fibrosis F3-4 should be reported and data on portal hypertension should be added if available. It is known that there is an increase of portal pressure in NASH patients also in non-cirrhotic patients.
This is a very important comment, especially because the risk of portal hypertension is usually overlooked in patients with colorectal liver metastases and non-cirrhotic liver. Not only NASH but also NRH may lead to portal hypertension.
In the present series, very few patients had F3-4 fibrosis (three patients, all F3, none had cirrhosis; none of them had NASH). No patient had clinical or radiological signs of portal hypertension. Please consider that a low platelet count can be related to oxaliplatin administration, making the non-invasive evaluation of portal hypertension more difficult. No patient had an endoscopy to assess esophageal varices. According to the surgical reports, no patient had evident signs of portal hypertension at surgery.
Even if we agree that NASH (and NRH) may lead to portal hypertension and, consequently, to increased operative risk, we did not have to manage such condition in the present series.
We added the data in the Results (lines 103-104) and a comment in the Discussion (lines 297-298).
Minor points:
- Please explain in extenso the abbreviations all over the paper (e.g. VOI, SinDil…)
We have now detailed the acronyms the first time they were used. For the explanation in extenso of VOI please see line 121, for SinDil please see line 41.
- I suggest to change “lobular flogosis” with “lobular inflammation” (of course “flogosis” is an elegant term and it has the same meaning but “inflammation” is the term commonly used in the definitions and guidelines and it would be better to align the terminology to the literature).
Thank you for the suggestion, the manuscript was corrected as requested.
- Table 1: it is visually not very clear what “grade 2-3” refers to. Please consider to reformulate.
Table 1 was corrected, and table legends were integrated to be clearer for the reader.
Reviewer 3 Report
Costa et al. present a paper titled “Virtual Biopsy for Diagnosis of Chemotherapy-Associated Liver Injuries and Steatohepatitis. A Combined Radiomic and 3 Clinical Model in Patients with Colorectal Liver Metastases.” The authors conclude that radiomics data, when combined with clinical and laboratory data, can improve the diagnostic accuracy of chemotherapy-related liver injury (CALI). They compared regions of the non-neoplastic liver from radiologic images (i.e., “virtual biopsies”) with the histopathologic features determined from colorectal metastasis resection specimens. The paper has presented some results/algorithms that could potentially be clinical useful. However, the manuscript could be substantially improved by the following changes:
- The authors have grouped the histopathologic features of oxaliplatin (FOLFOX)-based regimens with those from Irinotecan. However, it is well known in the literature that the histopathologic changes from these two agents are quite different. FOLFOX causes more severe sinusoidal injury with prominent sinusoidal dilatation and congestion (i.e., prominent NRH changes)( PMID: 14998849). In contrast, irinotecan causes more steatosis and steatohepatitis (PMID: 28053052). This should be more clearly explained in the introduction.
- In the material and methods, the authors should mention how far from the metastatic tumor lesions they are evaluating the non-neoplastic liver parenchyma. At least 1.0 cm from the masses should be required and some studies measure these at even greater distances (1.5 cm; PMID: 28061766) because it is well known that the tumor can cause mass-like effects, including NRH.
- How are the authors going to consider steatosis that developed in the liver from other risk factors like increased BMI, diabetes, etc.? The Tables that present these data are not complete. For example, why do tables 2, 3, and 4 have different variables? Table 4 needs to include BMI as a variable; why would you include it for NRH, but not for NASH? That makes no sense… ideally, you should include other variables that are common risk factors for NASH like hyperlipidemia, diabetes, alcohol use, etc. These tables should also include Table legends at the bottom of them that define the acronyms that are used, e.g., HUQ2 and GLZLM_f2. Not everyone is a radiologist, so these need to be clearly defined, especially if these are to be used clinically in the future. Other possible etiologies of NASH and the results of the above changes should also be mentioned in the discussion. Fatty liver disease is on the rise throughout the world and also in the U.S.
- The algorithms need to be explained better. The figure legends should explain the results and flow of these. I do not understand how to use them as they are presented. What do the numbers in the squares mean? What do the small numbers at the top of the squares mean? They are so small, I can barely read them.
- It would be nice to see a table that compares the differences in the histopathologic changes between the patients that received FOLFOX versus Irinotecan.
- The manuscript needs to be thoroughly reviewed for grammar, punctuation, and use of the English language. For example, flogosis should be replaced with the word inflammation.
Author Response
“Virtual biopsy for diagnosis of chemotherapy-associated liver injuries and steatohepatitis. A combined radiomic and clinical model in patients with colorectal liver metastases.”
Point-by-point response to reviewer #3
Review Report in bold
Author's Notes to Reviewer in italic
Costa et al. present a paper titled “Virtual Biopsy for Diagnosis of Chemotherapy-Associated Liver Injuries and Steatohepatitis. A Combined Radiomic and 3 Clinical Model in Patients with Colorectal Liver Metastases.” The authors conclude that radiomics data, when combined with clinical and laboratory data, can improve the diagnostic accuracy of chemotherapy-related liver injury (CALI). They compared regions of the non-neoplastic liver from radiologic images (i.e., “virtual biopsies”) with the histopathologic features determined from colorectal metastasis resection specimens. The paper has presented some results/algorithms that could potentially be clinical useful. However, the manuscript could be substantially improved by the following changes:
1. The authors have grouped the histopathologic features of oxaliplatin (FOLFOX)-based regimens with those from Irinotecan. However, it is well known in the literature that the histopathologic changes from these two agents are quite different. FOLFOX causes more severe sinusoidal injury with prominent sinusoidal dilatation and congestion (i.e., prominent NRH changes)( PMID: 14998849). In contrast, irinotecan causes more steatosis and steatohepatitis (PMID: 28053052). This should be more clearly explained in the introduction.
We thank the reviewer for his/her comment. This is an important point. In the introduction, we better clarified the association of different CALIs with specific chemotherapy regimens. Please see lines 70-71.
2. In the material and methods, the authors should mention how far from the metastatic tumor lesions they are evaluating the non-neoplastic liver parenchyma. At least 1.0 cm from the masses should be required and some studies measure these at even greater distances (1.5 cm; PMID: 28061766) because it is well known that the tumor can cause mass-like effects, including NRH.
Again, we thank the reviewer for this comment. The assessment of CALI was performed on the resected specimens. The assessment of CALI was done on a sample of non-tumoral liver parenchyma of at least 1 cm2. The sample was taken at a distance of at least 10 mm (20 mm whenever possible) from the tumor and the resection margin. This information was added in the Methods (lines 351-353).
3. How are the authors going to consider steatosis that developed in the liver from other risk factors like increased BMI, diabetes, etc.? The Tables that present these data are not complete. For example, why do tables 2, 3, and 4 have different variables? Table 4 needs to include BMI as a variable; why would you include it for NRH, but not for NASH? That makes no sense… ideally, you should include other variables that are common risk factors for NASH like hyperlipidemia, diabetes, alcohol use, etc.
This is a very important point. We agree that liver injuries can be associated with risk factors other than chemotherapy. The text was probably unclear in its previous version, but the analysis was carried out exactly as you suggested. The explorative analysis of the association between clinical variables and CALI (univariate analysis) is reported in table S1. It considered the same variables for all liver injuries, including metabolic syndrome, age, and BMI. Tables 2-3-4 report only the variables retained in the final predictive model at multivariate analysis. We specified this more clearly in the Methods (lines 363-378).
Of note, no patient of the present series had alcohol abuse. We also explored the association between chronic medications and liver injuries. We considered drugs that could induce NASH, NRH, or sinusoidal injuries according to the LiverTox NIH database. Two (3%) patients had chronic medications potentially associated with NASH (steroids in one, and tamoxifen in one). None of these patients had NASH at the final pathology evaluation. No patient had chronic medications potentially associated with sinusoidal dilatation or NRH. These data were added in the Results (lines 97-98 and 116-117).
These tables should also include Table legends at the bottom of them that define the acronyms that are used, e.g., HUQ2 and GLZLM_f2. Not everyone is a radiologist, so these need to be clearly defined, especially if these are to be used clinically in the future.
We added table legend defining all the radiomic variables used in the tables, as suggested.
Other possible etiologies of NASH and the results of the above changes should also be mentioned in the discussion. Fatty liver disease is on the rise throughout the world and also in the U.S.
In table S1 we reported the association of age, BMI, and metabolic syndrome with the different CALI. We analyzed the potential role of chronic therapies other than chemotherapy that could have blurred the histological picture, and no association was found. Please see my answer to your previous comment. We added these points in the Results (lines 96-98 and 115-117) and Discussion (line 299-301) sections, as suggested.
4. The algorithms need to be explained better. The figure legends should explain the results and flow of these.
In the revised version of the paper, we improved the legends of the Figures providing a wider amount of information, and making them more precise and self-explaining, to drive the reader to the process flow.
I do not understand how to use them as they are presented. What do the numbers in the squares mean? What do the small numbers at the top of the squares mean? They are so small, I can barely read them.
In the revised version of the paper, we devoted a strong effort to better explain the algorithms, how they work and which kind of evidence they provide. In particular, we added suitable comments in Figures legend, which hopefully make the interpretation of the Figures more reader-friendly. Moreover, we provided some details on how the classification tree-based method works at the end of Section 4 (lines 386-396): “A decision tree was built with the variables retained by the backward stepwise selection of the multivariate model to highlight and exploit the possible nonlinear association with the outcome. Indeed, a decision tree for classification problems is a top-down greedy algorithm that divides the predictor space into distinct and non-overlapping regions (identified by the criteria/split adopted to reach each node). The slitting criteria for decision rules are defined according to the principle of minimizing the variability of the response within each node. For every observation falling into one region (node), the decision tree predicts the occurrence of the corresponding CALI of interest. Decision trees are displayed as dendrograms to highlight decision steps. Each node of the tree reports: (a) the response mode class in the node, i.e., the predicted outcome for that node (presence of CALI =1; absence of CALI =0, the top number in the square); (b) the percentages of observations in the node belonging to the first response class (absence of CALI) and the second response class (presence of CALI) (the two central numbers in the square, summing up to 1); (c) the percentage of the total population falling into the node (the bottom number in the square). Decision rules are specified on each split.”
5. It would be nice to see a table that compares the differences in the histopathologic changes between the patients that received FOLFOX versus Irinotecan.
The association between different CALI and chemotherapy regimens is reported in table S1.
6. The manuscript needs to be thoroughly reviewed for grammar, punctuation, and use of the English language. For example, flogosis should be replaced with the word inflammation.
The manuscript was thoroughly reviewed for the English language.
Reviewer 4 Report
I read with interest this research paper, and the Authors should be commended for its originality.
I have one remark about the clinical significance: in which way the reader (or more in general clinicians) might integrate in clinical practice the results of this research? Based on the results presented, it seems that the topic is at its exploratory phase.
The second remark is about the conlcusions, for which I suggest the Authors to downgrade the strength of statements. Based on a retrsopective study, it is not possible to "demostrate" any kind of result. I would suggest to replace with something like "we observed".
Author Response
Response to the comments of Reviewer #4
I read with interest this research paper, and the Authors should be commended for its originality.
- I have one remark about the clinical significance: in which way the reader (or more in general clinicians) might integrate in clinical practice the results of this research? Based on the results presented, it seems that the topic is at its exploratory phase.
We thank the reviewer for his/her observation. The application of radiomics to clinical practice is still to be accomplished and remains the challenge of research in the near future. This is an exploratory phase. We think that our preliminary data can provide a basis for an innovative approach to the preoperative identification of liver injuries.
We better detailed these aspects in the discussion (lines 416-417 and 428-430) and in the conclusions (lines 439-441).
Nevertheless, even at this phase, some initial proposals can be advanced. For example, the identified cut-off value of HUQ2 for NASH diagnosis is a usable tool in clinical practice that needs for external validation.
- The second remark is about the conclusions, for which I suggest the Authors to downgrade the strength of statements. Based on a retrospective study, it is not possible to "demostrate" any kind of result. I would suggest to replace with something like "we observed".
We agree with the reviewer’s comment. We modified the conclusions as suggested, in order to mitigate the strength of our statements. We replaced the term “demonstrated” with “it was observed” (line 437). We also removed the adverb “consistently” in the last sentence.
Round 2
Reviewer 3 Report
No additional comments.
Author Response
Response to the comments of Reviewer #3
- Moderate English changes required
As requested by the reviewer, the manuscript has been revised by a native English speaker.
- No additional comments
We thank the reviewer for his/her review.